# Tribocorrosion and Surface Protection Technology of Titanium Alloys: A Review

**DOI:** 10.3390/ma17010065

**Published:** 2023-12-22

**Authors:** Yang Li, Zelong Zhou, Yongyong He

**Affiliations:** 1School of Nuclear Equipment and Nuclear Engineering, Yantai University, Yantai 264005, China; zhouzelong20000313@163.com; 2State Key Laboratory of Tribology, Tsinghua University, Beijing 100084, China

**Keywords:** titanium alloys, tribocorrosion, marine, plasma surface engineering

## Abstract

Titanium alloy has the advantages of high specific strength, good corrosion resistance, and biocompatibility and is widely used in marine equipment, biomedicine, aerospace, and other fields. However, the application of titanium alloy in special working conditions shows some shortcomings, such as low hardness and poor wear resistance, which seriously affect the long life and safe and reliable service of the structural parts. Tribocorrosion has been one of the research hotspots in the field of tribology in recent years, and it is one of the essential factors affecting the application of passivated metal in corrosive environments. In this work, the characteristics of the marine and human environments and their critical tribological problems are analyzed, and the research connotation of tribocorrosion of titanium alloy is expounded. The research status of surface protection technology for titanium alloy in marine and biological environments is reviewed, and the development direction and trends in surface engineering of titanium alloy are prospected.

## 1. Introduction

Titanium alloy has the advantages of low density, high specific strength, good corrosion resistance, good heat resistance, low-temperature resistance, being non-magnetic, good impact resistance, and good welding performance [1,2,3,4,5,6,7,8]. It is widely used in aerospace, marine engineering, medical biology, and other fields [9,10,11,12,13,14,15]. In ship and marine engineering, titanium and titanium alloys have better seawater corrosion resistance than stainless steel and copper alloys [16,17,18]. They are widely used in ship hulls, sea pipelines, pumps, valves, seawater desalination devices, deep-sea detectors, offshore oil platforms, etc., known as ‘marine metals’ [19,20]. In addition to being corroded by seawater, titanium and titanium alloy structural materials are also subjected to mechanical effects such as friction/erosion, often leading to unpredictable sudden failure and causing huge losses. Therefore, studying their tribocorrosion behavior and mechanism is necessary.

Titanium alloy has developed into a primary material for high-end surgical implants with high specific strength, good biocompatibility, and mechanical compatibility [21,22,23,24,25]. In particular, additive manufacturing technology based on a digital model can form a personalized titanium alloy with a porous structure, which can effectively solve the problem of stress shielding between dense metal and human bone and is conducive to the growth of bone tissue [26,27,28,29,30,31,32,33]. Titanium alloy implants are more and more widely studied and applied. The global medical device market is expected to grow from nearly USD455 billion in 2021 to USD658 billion in 2028 due to increasing healthcare facilities, the elderly population, healthcare expenditures, and technological advances [34].

Tribocorrosion in the marine environment can cause damage to titanium alloy components, affecting their service life and safety [35,36,37]. The synergistic effect of friction and corrosion is mainly manifested as physical spalling and chemical corrosion on the surface of titanium alloy [38,39,40]. The physical spalling is mainly caused by the impact of seawater and the wear of particles. At the same time, chemical substances in seawater, such as oxidants, sulfides, and chlorides, will rapidly erode the surface of titanium alloys and accelerate their corrosion rate [41,42,43,44]. When physical exfoliation and chemical corrosion exist simultaneously, the synergistic effect of tribocorrosion will be formed, accelerating the damage to titanium alloy parts [45,46].

When titanium alloy is used as an implant material in a complex human environment, the implanted titanium alloy will be corroded by the surrounding body fluid, and the bone interface between the implant and the tooth will be worn [3,47,48,49]. Under the synergistic effect of tribocorrosion, the implanted titanium alloy will be damaged due to the decrease in fatigue, and the wear debris and corrosion products generated during the tribocorrosion process will lead to cell damage and cause inflammation, allergic reactions, and rashes, which will cause significant harm to health [50,51,52,53,54].

In summary, the prevention of tribocorrosion of titanium alloy in the human environment and marine environment is of great significance to ensure the safety of use, reduce maintenance costs, and prolong the service life of components. Various surface modification techniques have been widely used to improve tribocorrosion performance—for example, PVD, plasma spot oxidation, plasma chemical heat treatment, laser cladding, etc. Figure 1 shows the application scenarios of titanium alloys and some surface modification techniques and technical principles used to improve titanium alloys’ friction and corrosion properties.

With the continuous expansion of the application range of titanium alloy in marine engineering, biomedicine, and other fields, it is necessary to study the tribocorrosion problems of titanium alloy under special working conditions. Surface modification of titanium alloy can effectively improve its tribological properties in seawater environments and human body fluids, prolong its service life, and improve its reliability.

This paper reviews the research background of tribocorrosion of titanium alloys, focusing on the marine and human environments. Subsequently, various surface modification methods for improving the tribocorrosion properties of titanium alloys are introduced. Finally, the research on friction corrosion of titanium alloy is prospected.

## 2. Titanium Alloy and Its Classification and Properties

Titanium is an allotrope element with different crystal shapes. Titanium alloy has a hexagonal close-packed (hcp) crystal structure at room temperature, labeled as α phase [59]. The structure is transformed into a ‘body-centered cubic’ (bcc) crystal structure at 883 °C, labeled as the β phase [59,60,61,62]. This temperature is called ‘β-trans’, which refers to the temperature transition from α phase or α-β phase to β phase. β-trans is the lowest temperature at 100% β [59].

Titanium alloys are divided into non-alloyed titanium, α, β, and α-β titanium alloys [59,63,64,65]. Unalloyed titanium can be divided into grades 1, 2, 3, and 4, which can be used as implants. The α alloy has a complete α structure. The β titanium alloy can be quenched into cold water on its β cross-section, and the β phase has no decomposition of martensite. At room temperature, α-β alloy is the combination of β phase and α phase. The alloying elements in titanium alloy are divided into α stabilizer and β stabilizer [66,67]. Alpha stabilizers, such as aluminum and oxygen, can increase the temperature at which the alpha phase is stable. β stabilizers, such as vanadium and molybdenum, are responsible for the constancy of the β phase at lower temperatures [68,69]. Table 1 shows some common types of titanium alloys and their mechanical energy, including elastic modulus, yield strength, and tensile strength.

## 3. Tribocorrosion Behavior

Tribocorrosion behavior usually refers to the damage phenomenon that occurs under the action of friction motion and a corrosive medium [70,71,72]. Compared with wear damage, the movement of the contact surface in the process of tribocorrosion not only directly wears the material but also destroys the protective film so that the surface of the fresh material continuously contacts the corrosive fluid to accelerate the corrosion, forming an interaction between corrosion and wear [73,74,75]. Therefore, coupled tribocorrosion damage is a superposition of tribocorrosion damage and a significant increase in material loss due to corrosion, wear, and their interaction [76,77,78,79,80].

Tribocorrosion is produced under the combined action of corrosion and wear, so many severe influencing factors exist. The main elements are the medium, material, working conditions, and electrochemical characteristics (Figure 2). The typical factors are [81,82,83,84,85]: 

Corrosive medium

The factors that affect the corrosion performance of the material, such as the pH value, medium concentration, and temperature, also influence the tribocorrosion performance of the material. On the one hand, the corrosive medium directly produces corrosion. On the other hand, the medium can react to form new corrosive substances and increase the corrosion wear of the material.

Wear condition

Similar to traditional tribology, the tribocorrosion properties of materials are closely related to the contact mode (such as point, line, or surface contact), motion mode (such as rolling, sliding, or vibration), load, speed (frequency), and other factors between friction pairs. The results show that the speed and load have the most significant influence on the tribocorrosion properties of the passivation metal, which not only affects the structure and properties of the friction contact surface of the passivation metal but also affects the formation and destruction of the passivation film.

Material chemical composition and microstructure properties

The main factors affecting the tribocorrosion resistance of materials are the chemical composition, microstructure, hardness, plasticity, and surface roughness. Therefore, wear-resistant and corrosion-resistant integrated materials need to have excellent tribocorrosion resistance, which needs to be comprehensively optimized from chemical composition and microstructure aspects. The structure and mechanical properties of the material are the decisive factors determining its tribocorrosion properties. Only when the material has good tribocorrosion resistance can it be applied to a corrosive wear environment.

Electrochemical factors

Electrochemical factors include the applied potential, formation, and destruction of passivation film, etc., which are also external factors affecting the tribocorrosion of materials.

To quantify the effect of tribocorrosion on the whole tribocorrosion process, Annsley Mace et al. [87] measured each influencing factor (V_T_, V_P_, V_I_, and V_ICPMS_) using various techniques. Figure 3 shows a schematic of the overall process of tribocorrosion. They believe that the total volume of the material ‘removed’ is a groove (V_tr_) caused by the cyclic sliding of a single diamond in the metal, including the generation of plastically deformed metals and wear particle fragments (metals and oxides) on the surface and below. Secondly, there are accumulated materials around the groove. Ideally, since plastic deformation is an equal-volume process, the amount of track (V_P_) produced by plastic deformation should be similar to the volume of accumulated plastic deformation materials. The V_ICPMS_ was obtained by measuring the metal ions and fine suspended particles in the solution using ICP-MS. Finally, the instantaneous measurement of the fretting corrosion current during the fretting process provides a method to determine the charge associated with the passivation reaction (i.e., oxide formation) and any anodic ion dissolution reaction and is related to the current-based volume (V_I_). The volume balance analysis compares the importance of the groove with other volume definitions. Ideally, the two volumes will be equal, according to: V_tr_ = V_pl_ + V_m_ + 1/PB·V_ox_ + V_ion_(1)

The amount that can be measured in the experiment has the following relationship:V_p_ = V_pl_ + V_m_ + V_ox_(2)

Among them, V_TR_ is the volume of the cell, V_pl_ is the volume based on plastic deformation in the accumulation, V_m_ is the volume of the metal particles related to the volume of the collection, V_ox_ is the volume of the generated oxide fragments, and Vion is the volume of the metal released into the solution as an ion. The constant PB is the Bedworth ratio of Ti.

According to two Formulas (1) and (2), and introducing the sum of the tribological volume and the corrosion volume (V_Tc_), consider the volume of ions released into the solution V_ICPMS_; the following relationship is finally obtained to characterize the total friction corrosion volume loss: V_Tc_ + V_c_ + V_wear_ = V_p_ + V_I_ (1 − PB) + V_ICPMS_·PB(3)

Assuming that the measured charge directly produces metal oxides (mainly TiO_2_), V_I_ can be measured using constant potential measurement and estimated according to Faraday’s law.

The steps and equipment of tribocorrosion characterization can refer to the research of Mathew et al. [88]. They first performed basic electrochemical tests using a standard three-electrode setup (Figure 4a). A saturated calomel electrode (SCE) was used as the reference electrode, a graphite rod was used as the auxiliary electrode, and the exposed surface of Ti was used as the working electrode. The potentiometer was connected to the computer for data acquisition for corrosion measurement. The open circuit potential (OCP) was initially monitored within 3600 s to assess the potential and stabilize the system. In addition, the sample was circularly polarized from −0.8 V to 1.8 V at a scan rate of 2 mV/s. The friction corrosion test equipment is shown in Figure 4b. The tribological system consists of an alumina ceramic ball rubbed against a Ti sample (pin with a flat surface) in an electrolyte chamber. The standard three-electrode model shown in the basic electrochemical test was used. The friction corrosion test adopted the classic scheme (Figure 4c). Initially, the sample was electrochemically cleaned at a cathodic potential of −0.9 V_vs. SCE_. This was carried out to achieve the same start-up conditions. The OCP was then monitored within 600 s to evaluate the potential and stabilize the system (using a newly formed TiO_2_ film). Next, electrochemical impedance spectroscopy (EIS) tests were performed to study the properties of the oxide film formed on the Ti surface (corrosion kinetics). The frequency range of EIS measurement was 100 kHz to 5 MHz, and the amplitude of the AC sine wave was ten mV, oscillating near the corrosion potential. The potential and current were measured to determine the fundamental (Z′) and imaginary (Z″) components of the impedance, plotted with a Nyquist plot, or the total impedance (|Z|) and phase angle.

## 4. Tribocorrosion Behavior of Titanium Alloy

### 4.1. Tribocorrosion Behavior of Titanium Alloys in the Marine Environment

There are a variety of inorganic salts dominated by NaCl in seawater, dissolved oxygen, particulate organic matter, and humic substances, including humic acid. O_2_ and CO_2_ in surface seawater are near saturation, and the pH is about 8.2. It is a corrosive electrolyte solution with a very complex composition. Corrosion-accelerated wear is mainly manifested as Cl^−^ leads to brittle spalling of the material surface, increases corrosion activity, destroys the passivation film, and promotes wear [89]. The wear-accelerated corrosion is mainly manifested in the removal of surface corrosion products by friction and wear, the destruction of surface passivation, and Cl^−^ hindering the repair of the passivation film. The formation of electrochemical primary cells around the wear scar accelerates the corrosion. Wear also leads to an increase in the specific surface area of the wear surface, resulting in plastic deformation, so that the surface is in a high-energy reactive state and promotes the occurrence of corrosion [90,91].

The synergistic interaction between tribocorrosion in corrosion wear of metals or alloys is usually positively correlated, which promotes each other and accelerates corrosion wear. Zhang et al. [92] compared the fretting wear mechanism and characteristics of TC4 titanium alloy in pure water and a 3.5% NaCl (mass fraction) solution. They found that the process of corrosion, wear, corrosion, and wear showed a ‘positive interaction’ relationship; that is, the interaction between corruption and wear aggravated the loss of materials. Chen et al. [93] found that the wear amount of Ti6Al4V in seawater was significantly greater than that in pure water when paired with Al_2_O_3_ in purified water and simulated seawater, indicating that corrosion accelerated wear.

Pejaković et al. [45] studied the tribo-electrochemical properties of Ti6Al4V under three contact loads (10 mN, 100 mN, and 1 N) in artificial seawater. The results suggest a wear mechanism, as shown in Figure 5. At the lowest load (10 mN), the tangential force in the sliding direction is not strong enough to remove the passivation film, resulting in sliding occurring at the top of the asperities without wearing the film, leading to a lack of wear. Under the highest load, the tangential force is large enough to cause the passivation layer to break and cause significant wear, exposing the metal to a corrosive environment. This is a common situation reported in friction corrosion experiments using point contact. The contact pressure increases sharply under the typical normal load in the Newton range. Under an intermediate load of 100 mN, the average load is assumed to reach a critical value, resulting in the removal of the convex part, which leads to the partial passivation of the metal observed in the friction corrosion experiment and the presence of oxide fragments in the wear traces.

Ding et al. [94] studied the fretting wear characteristics of TC11 titanium alloy in pure water and simulated seawater. The material loss in seawater is always less than that in water, and the tribocorrosion is a negative interaction. The analysis shows that under fretting conditions, the film produced by active components such as sulfur, phosphorus, and chlorine in seawater plays a role in reducing friction and controlling wear, preventing the generation of a large number of abrasive particles, reducing or even eliminating the damage caused by ‘micro-cutting’ and ‘plowing’ so that corrosion shows a negative interaction in wear.

Xv et al. [95] studied a novel approach β and α + β Friction corrosion behavior of bi-omedical titanium alloy in bovine serum solution under open circuit potential. At slower sliding speeds, the alloy has the ability to re passivate. The results show that re pas-sivation is mainly related to the formation of a larger friction film on the worn surface. The friction layer, as a solid lubricating film, stabilizes the friction coefficient. Figure 6 shows the layered structure on the surface of the sample.

There is no simple tribocorrosion between metal motion pairs in the marine environment. Scholars usually divide the interaction with tribocorrosion into two categories: positive and negative interactions with tribocorrosion. The positive interactions can be divided into the promoting effect of corrosion on wear and the promoting effect of wear on corrosion. Similarly, the negative interactions can be divided into the inhibition of corrosion on wear and the inhibition of wear on corrosion. Moreover, positive and negative interactions are not immutable; they will interact, transit, and transfer under different materials, working conditions, and environmental conditions.

### 4.2. Tribocorrosion Behavior of Titanium Alloys in the Human Body Environment

Titanium alloy has been used in joint replacement, bone fixation, dental implants, cardiac pacemakers, artificial heart valves, stents, and high-speed blood centrifuges [27,96,97,98,99,100]. However, the elastic modulus of the medical titanium alloys currently used is as high as 90~115 GPa, which is much higher than that of human cortical hard bone (10~25 GPa) and cancellous bone or cartilage (0.05~3 GPa) [101,102,103,104,105]. Such a significant difference in elastic modulus will lead to an unbalanced stress distribution between the implant and the surrounding bone tissue, resulting in stress shielding and loosening of the implant relative to the bone tissue [105,106,107,108,109,110,111,112].

Porous titanium alloy materials have an elastic modulus matching that of human bone tissue, effectively solving the elastic mismatch between implants and human bone [113,114,115,116,117,118,119]. Recently, additive manufacturing technology has been used to prepare porous titanium alloy materials [120,121,122,123,124,125]. This method uses tool software such as 3DXpert (France) and Simulate Additive (Sweden) to design the absorbent structure. Then, the computer controls the laser beam/electron beam layer-by-layer melting process according to the program and obtains a porous material entirely consistent with the expected structure [126,127]. Three-dimensional printing technology has the advantages of precise digital preparation, high efficiency, a short cycle, and customization [128,129,130]. It benefits preparing medical metal porous implant devices with complex structures. The large number of pores inside the porous material is more conducive to the growth of surrounding cells and the growth of new bone, thus significantly promoting the ability of bone tissue formation [131,132,133]. Therefore, porous titanium alloy implant devices have become a research hotspot in metal implants. However, due to the high melting point of titanium alloy and its good affinity with O_2_ and N_2_ in the air at high temperatures, it is challenging to prepare porous titanium alloy using the liquid foaming method.

Although under normal conditions, the surface of titanium alloy will generate a very stable and continuous oxide passivation film with solid bonding, the passivation film may be peeled off and dissolved under the erosion of external forces and body fluids due to the complexity of the human environment. Therefore, substances will still be released into the tissue during use. In addition, the friction coefficient of titanium alloy is significant and the wear resistance is poor, resulting in a large amount of Ti, Al, and V black debris due to wear after implantation, causing aseptic loosening and eventually leading to joint replacement failure. In addition, Al and V elements have potential cytotoxicity, which may lead to the inability to form apatite on the surface, especially Al, which is may cause Alzheimer’s disease. Figure 7 shows the characteristic severe corrosion images of the total hip prosthesis components and the recycled Ti6Al4V conical samples [134].

When titanium alloy is used as a dental implant in a complex oral environment, the implant will be subjected to different forces during the chewing process, resulting in mic-romotion at multiple interfaces, forming a tribocorrosion system. Vieira et al. [138] studied the effects of varying pH values and corrosion inhibitors on pure titanium during tribocorrosion in artificial saliva. They showed that the tribocorrosion properties are slightly improved after adding citric acid/anode corrosion inhibitor. The reason may be that the redox reaction occurs in the contact area during the fretting process, forming a passivation layer in the contact area. The hardness of dental implant materials is significantly higher than Young’s modulus of bone. The unevenness of relatively rigid materials will eventually penetrate the surface of rela-tively soft materials, a phenomenon known as microplowing. In Figure 8, the abrasive particles in each movement form a plow, forming a symmetrical furrow. The subsequent groove is parallel to the direction of the abrasive in motion. The proximity of this phenomenon is not conducive to relatively soft materials, causing substantial local deformation and material removal [139].

## 5. Surface Modification Methods

Surface strengthening treatment is an effective technical means to improve the tribocorrosion resistance of titanium alloy. Many surface treatment technologies are applicable for enhancing the feeling of titanium alloy [27,142,143,144]. The application and development of titanium surface treatment technology have gone through three stages: The first stage is the traditional surface technology represented by electroplating, electroless plating, and thermal diffusion (nitriding, carburizing, etc.). The second stage is the modern surface treatment technology marked by plasma and electron-beam coating technology, laser surface strengthening, thermal spraying, and micro-arc oxidation technology. The third stage is the composite application of various surface treatment technologies. The multi-element, multilayer, gradient structure, and ultra-thick design and preparation of the surface modification layer meet the surface protection requirements of titanium metals in high-speed, heavy-load, and harsh complex-media environments [80,145,146].

Compared with the development of new materials, surface modification has significant advantages and operability in terms of cost. The modification technology of medical titanium alloy is mainly divided into dry amendment and wet modification [1,146]. Dry modification technology carries out various chemical reactions or thin-film deposition on the surface of the titanium alloy substrate in the gas phase, such as magnetron sputtering, vapor deposition, ion implantation, laser cladding, ion beam sputtering, and so on. Wet modification technology is mainly a chemical reaction between the titanium alloy matrix and the elements in the solution environment to achieve surface modification, such as electrochemical deposition, hydrothermal synthesis, and other methods. Table 2 summarizes some surface modification methods and their effects on tribocorrosion properties:

### 5.1. Chemical Heat Treatment

Chemical heat treatment is a process in which the material is placed in a medium containing a particular active element for heating and heat preservation, and a concentration gradient is formed on the surface to make the active ingredient enter the surface of the material to form a layer [164,165,166,167]. Chemical heat treatment can effectively improve the thermal fatigue resistance, wear resistance, and corrosion resistance of materials, and chemical heat treatment has a low cost, wide application, and mature technology [168,169,170,171,172,173,174,175]. After chemical heat treatment, the surface composition of the material will change [176,177]. After chemical heat treatment, the material surface’s hardness, strength, wear, and corrosion resistance are greatly improved [178,179,180,181,182].

The chemical surface heat treatment methods mainly include nitriding, carburizing, oxidating, and boronizing. Nitriding treatment is the most commonly used chemical heat treatment method for titanium metal surfaces, including gas nitriding, plasma nitriding, and laser nitriding [183,184,185,186,187,188,189,190,191,192]. The nitriding temperature of titanium metal is much higher than that of steel materials. It must be above 800 °C to obtain a nitride layer with sufficient depth. The nitrogen element infiltrated into the surface of titanium metal forms a nitrogen and titanium solid solution of α phase and α + β phase, and a thin titanium nitride layer is formed on the outermost layer. The thickness of the hardened layer obtained with a nitriding treatment is generally not more than 200 μm, and the hardness of the nitrided layer is about 10 GPa.

Zhao et al. [193] indicated that plasma nitriding on pure Ti produces a thick, dense, and homogeneous TiN layer similar to DLC coating. This provides a comparatively better performance as an implant biomaterial, since it exhibited excellent electrochemical, tribological, and tribocorrosion performance. Wang et al. [180] studied the effect of plasma nitriding and TiN coating dual-phase treatment on the corrosion resistance of cast titanium. Compared with the untreated titanium, the dual-phase-treated titanium showed a higher corrosion potential (E_corr_) and lower corrosion current density (I_corr_). The SEM results showed that the surface corrosion of untreated titanium was more severe than that of dual-phase-treated titanium. Guan et al. [194] used a self-made induction nitriding device to nitride medical pure titanium (TA1) and used Hank’s solution to test its wear resistance and corrosion behavior. The tribocorrosion behavior and synergistic effect were studied with electrochemical measurements and surface morphology analysis. The experimental results in Figure 9 showed that a nitrided layer of about 20 μm thick could be obtained after one hour of induction nitriding treatment of TA1, which effectively improved the mechanical properties, corrosion resistance, and wear resistance of TA1. The pure wear weight loss and pure corrosion weight loss of nitrided samples were one order of magnitude lower than those of untreated samples. The synergistic effect was produced in friction corrosion, and weight loss caused by wear was dominant. However, the temperature of the chemical surface heat treatment was high, and the treatment time was extended. The high temperature and long-duration chemical heat treatment can easily affect the fatigue performance of titanium metal, and the corrosion resistance was reduced.

Thermal oxidation refers to exposing the material to an oxygen atmosphere in a high-temperature environment to cause an oxidation reaction on the surface [195,196,197]. Thermal oxidation is usually used to improve the material surface’s heat resistance, corrosion resistance, and oxidation stability. By reacting with oxygen at high temperatures, an oxide layer is formed on the material’s character, which can increase its surface hardness and reduce the surface friction coefficient. In thermal oxidation, metal materials react with oxygen to produce metal oxides. These oxides form a dense, uniform oxide layer that can adhere to the metal surface. The thickness, composition, and properties of the [198] oxide layer depend on the temperature, oxygen concentration, and thermal oxidation time [75]. The oxide layer has good heat resistance and oxidation stability, which can protect the metal surface from high-temperature oxidation. Secondly, the oxide layer can improve metal materials’ corrosion resistance, reduce metals’ reaction with external gas and liquid, and prolong their service life.

R. Bailey et al. [199] studied the corrosion and tribocorrosion properties of thermally oxidized commercially pure titanium in a 0.9% NaCl solution. This treatment produces a multilayer structure consisting of a 1 μm rutile (TiO_2_) film and a 9 μm α titanium oxygen diffusion zone (ODZ) (α-Ti(O)). Compared with the untreated CP-Ti, the rutile oxide layer has low friction and better resistance to material removal during tribocorrosion. Four friction zones can be identified in the typical friction curve during the sliding wear under open circuit potential. Each friction zone has characteristics corresponding to the gradual or partial removal of the oxide layer, oxide layer, diffusion zone, and matrix. The abnormal anodic protection behavior of the oxide film is also observed. When titanium is positively polarized during sliding, the durability of the oxide layer is extended, thereby reducing friction and significantly reducing material loss. Cheraghali et al. [200] generated a more biologically active porous anodic oxide layer on the thermal oxide layer with ODL anodic oxidation technology. The effects of the average load and sliding distance on the tribocorrosion behavior of the anodic oxide layer produced using the ODL anodic oxidation process were studied. The results showed that the anodic oxide layer had sufficient durability at sliding distances of 5 m and 25 m and could resist friction stress. However, a further increase in the sliding distance leads to local damage to the oxide layer, which is then completely removed at a sliding distance of 300 m, especially under high normal loads of 1 and 1.5 N. ODL showed the lowest tribocorrosion rate, followed by anodic oxidation at 150 V and 175 V.

Si et al. [201] prepared a TiO_2_/SrTiO_3_ heterostructure coating on the surface of the Ti6Al4V alloy using thermal oxidation. The preparation process and mechanism of TiO_2_/SrTiO_3_ coating are shown in Figure 10. The effective removal of bacterial biofilm could be achieved under ultraviolet light irradiation. In addition, the tribocorrosion properties of TiO_2_/SrTiO_3_ coatings in SBF solution were also tested. The results showed that a thermal oxide layer with sufficient thickness provides a reliable tribocorrosion protection barrier. The thermal oxide layer was subjected to a periodic contact load and surface fatigue cracks. The wear scar pit was shallow, and the wear debris peeled off from the matrix as scales, a typical fatigue wear characteristic.

### 5.2. Plasma Electrolytic Oxidation

Plasma electrolytic oxidation (PEO), also known as micro-arc oxidation (MAO), is a surface treatment technology [202,203,204,205]. Metal materials are placed in the electrolyte, and a micro-arc discharge is generated by applying a high voltage, thus forming an oxide film on the metal surface. This oxide film usually has the characteristics of uniformity, density, and high hardness, which can provide good protection and improve the surface properties of metal materials [206,207,208,209]. The PEO process includes electrochemical oxidation, plasma chemical reaction, and thermal diffusion in the electrolyte. Micro-arc oxidation technology can grow a layer of coating in situ with high hardness and wear and corrosion resistance on the surface of valve metals such as titanium, which can significantly improve the surface properties of light metals. In addition, PEO can adjust surface properties such as hardness, elastic modulus, wettability, porosity, roughness, chemical composition, and crystallinity [210,211,212,213,214]. 

Compared with traditional surface treatment technologies such as anodic oxidation and electroplating, it is an environmentally friendly surface treatment technology with a short process flow and no heavy metal ions in the solution. In addition, the micro-arc oxidation process is stable and reliable, and the equipment is simple. The reaction is carried out at room temperature, which is easy to operate and master. The ceramic film is in situ grown on the substrate, the bonding is firm, and the ceramic film is dense and uniform.

S.A. Alves et al. [215] produced an anodic oxide film on the surface of cp-Ti with plasma electrolytic oxidation (PEO) technology. After PEO treatment, oxide films with varying surface characteristics, such as surface roughness, morphology, and film thickness, were achieved based on the anodic oxidation parameters. To examine the impact of PEO treatment on the tribocorrosion behavior, untreated and PEO-treated Ti samples were subjected to a reciprocating sliding test in artificial saliva. The results showed that PEO treatment effectively enhanced the resistance of the cp-Ti substrate to both electrochemical and mechanical influences, and the efficacy was influenced by the specific PEO process parameters employed. Laurindo et al. [216] evaluated the effect of PEO voltage and annealing treatment on tribocorrosion properties. The results showed that the PEO layer significantly improves the tribocorrosion resistance of bare titanium. A rutile phase also increases friction corrosion resistance through voltage or post-heat treatment. Sukuroglu et al. [217] used different frequencies to grow TiO_2_ coatings on Cp-Ti substrates, and the treatment method was PEO. The tribocorrosion properties of the coating were studied with a pin-on-disk wear test and potentiodynamic polarization test device. The effect of frequency change on the performance of the PEO coating was examined at a constant voltage. As a result, the increase in frequency lead to smaller pores and cracks in the surface morphology of the coating, which lead to an increase in the friction and corrosion behavior of the layer.

He et al. [218], in situ, synthesized Cu_x_O (CuO and Cu_2_O) on the ceramic coating on the titanium alloy substrate with plasma electrolytic oxidation (PEO) and systematically studied its biological activity and tribocorrosion behavior. The specific results are shown in Figure 11. During the tribocorrosion process, the formation of hydroxyapatite and CuO on the worn surface prevented direct contact between the coating and the counterpart, resulting in a practical lubrication effect, which reduced the mass loss of the prepared layer by 53.5% compared with the traditional PEO coating. More importantly, the sliding process accelerated the enrichment of Cu on the worn surface, and hydroxyapatite deposition on the dull surface induced the conversion of Cu_2_O to CuO. This behavior produces lubrication and prevents direct contact between the coating and the counterpart. It is worth noting that tribocorrosion is beneficial to long-term antibacterial activity, and a high surface Cu content enhances the long-term antibacterial activity of the Cu_x_O/TiO_2_ coating, which provides essential value for the clinical application of long-term sterilization of implant materials.

The micro-arc oxidation technology was developed based on anodic oxidation [204]. The process has outstanding characteristics: (1) The electrolyte is weakly alkaline and does not pollute the environment. (2) The process is simple. The pretreatment of the workpiece only requires oil removal and decontamination on the surface. It does not need to remove the natural oxide layer on the surface, which is suitable for large-scale automation. (3) Micro-arc oxidation can be completed at once or several times, and anodic oxidation must be restarted once interrupted. (4) No vacuum or low-temperature conditions are required. However, the formation process of micro-arc oxidation films is quite complex, and research on the mechanism is insufficient; the oxidation voltage is much higher than that of conventional anodic oxidation, and safety should be paid attention to during operation. The current efficiency of micro-arc oxidation is low; the electrolyte temperature rises rapidly and needs to be cooled.

### 5.3. Physical Vapor Deposition

Physical vapor deposition (PVD) technology refers to the use of physical methods to vaporize the surface of the material source (solid or liquid) into gaseous atoms or molecules under vacuum conditions or partially ionized into ions, and through a low-pressure gas (or plasma) process, a film with certain special functions is deposited on the surface of the substrate [219,220,221]. Physical vapor deposition is one of the leading surface treatment technologies. PVD coating technology is divided into three categories: vacuum evaporation, sputtering, and ion coating [222,223]. The main methods of physical vapor deposition are vacuum evaporation, magnetron sputtering coating, arc plasma coating, ion plating, and molecular beam epitaxy. The corresponding vacuum coating equipment includes a vacuum evaporation machine, a vacuum sputtering coating machine, and a vacuum ion coating machine. With advancements in deposition methods and technologies, physical vapor deposition technology has the capability to deposit various types of films including metal films, alloy films, compounds, ceramics, semiconductors, polymer films, etc. By employing various target materials, reaction gases, process methods, and parameters, nitriding enables the production of diverse coatings catering to different needs [224,225,226,227,228]. These coatings include surface-strengthened coatings with exceptional hardness and wear resistance, coatings with high density and chemical stability providing corrosion resistance, solid lubrication coatings, coatings in a wide range of decorative colors, as well as unique functional coatings used in electronics, optics, energy science, and other fields [56,229,230,231,232]. 

Plasma-enhanced magnetron sputtering (PEMS) is a method to improve magnetron sputtering technology using the plasma enhancement effect [233,234,235]. In the traditional magnetron sputtering process, the surface of the sputtering target is bombarded with ions, resulting in the shedding of the sputtering material. In plasma-enhanced magnetron sputtering, by generating plasma in the sputtering chamber and introducing energetic ions, the ion density and energy can be increased, thereby improving the quality and growth rate of the film [236]. Hatem et al. [148] synthesized a low-friction Ti-Si-C-N nanocomposite coating on ASTM F136 (Ti-6Al-4V ELI) titanium alloy using plasma-enhanced magnetron sputtering (PEMS) technology. The results showed that the wear rate of the low-friction Ti-Si-C-N nanocomposite coating was reduced by at least 97% compared with the bare titanium alloy sample. The tribocorrosion performance of the coating samples strongly correlated with the silicon and carbon content in the chemical composition of the coating. Therefore, the refinement of nanocrystals, the existence of amorphous carbon regions, and the generation of oxides are the related characteristics to improve Ti-Si-C-N nanocomposites’ friction and corrosion behavior. 

Unbalanced magnetron sputtering is a magnetron sputtering technology that is characterized by the introduction of a non-uniform magnetic field in the magnetron sputtering system to improve the spatial distribution of the electron cloud during the sputtering process, thereby improving the quality and performance of the film [237,238,239,240]. Fu et al. [241] prepared CrMoSiN coatings with different molybdenum contents on silicon and titanium alloy substrates with an unbalanced magnetron sputtering system. They tested the tribocorrosion behavior of the coatings sliding with SiC balls in seawater. After the addition of the Mo element to the CrSiN coating, (Cr, Mo) N replaces the solid solution and combines with CrN, Mo_2_N phase, and SiNx amorphous to form a nanocomposite structure in the CrMoSiN coating, and the columnar structure is dense. Therefore, the coating surface changes from a ‘fine dome’ shape to a relatively flat profile with a ‘groove boundary’. The columnar structure of the CrSiN coating is loose, and severe coating peeling occurs on its wear track. Under the synergistic effect of tribocorrosion, the material loss of the CrSiN coating begins to accelerate, as shown in the figure. For CrMoSiN coatings, due to the formation of the MoO_3_ layer (Figure 12) under polarization and wear behavior, the wear track of all CrMoSiN coatings becomes smooth without coating peeling marks. The three-body wear is eliminated, and the phenomenon of coating peeling is eliminated.

Radio frequency sputtering is a sputtering technology using a radio frequency power supply. It is a physical vapor deposition technology used to form a uniform, dense, high-quality film on the surface of the film material [242,243,244]. The main working principle of radio frequency sputtering technology is to introduce an inert gas into the sputtering device and make it collide with the atoms or molecules of the sputtering source by accelerating and bombarding the static gas ions. The atoms or molecules of these sputtering sources will be knocked out and form a film on the surface of the substrate material [245,246].

Çaha et al. [155] used sputtering technology to deposit TiN coating on the surface of cp-T to improve its tribocorrosion performance. Figure 13 shows SEM images of the worn scratches and worn sub surfaces, as well as the wear mechanisms of different samples. It was found that the uncoated and 30 min samples were mainly characterized by a combination of abrasive and adhesive wear, which was controlled by parallel grooves, discontinuous friction layers, plastic deformation, and transfer materials. In comparison, the abrasive and sticky wear characteristics of 80 min samples only occurred on the prominent surface. The reason was that with the increase in deposition time, the proportion of the Ti_2_N phase increases, and the corresponding coating exhibits a lower corrosion rate and better capacitance behavior, making it resistant to abrasive wear, adhesive wear, and plastic wear.

Zhao et al. [247] successfully prepared TiN coating and Ti/TiN multilayer coating on the surface of Ti6Al4V alloy using multi-arc ion plating technology. The corresponding characterization and wear mechanism models of different coatings are shown in Figure 14. The Ti/TiN multilayer coating has good friction corrosion resistance and a low friction coefficient. For the TiN coating, the surface compressive stress causes the crack to diffuse below the coating surface, accelerating the coating failure. The soft Ti layer in the Ti/TiN multilayer coating can effectively prevent the generation and propagation of cracks and interrupt the continuous growth of columnar crystals, making the coating more dense and the corrosive medium not effortless to penetrate the substrate. In the simulated body fluid environment at 37 °C, the tribocorrosion resistance of the multilayer coating is nearly twice as high as that of Ti6Al4V alloy and TiN coating. They concluded that the material loss caused by tribocorrosion is not the sum of superficial, pure corrosion, and pure wear loss, and the synergistic effect between tribocorrosion accounts for a large proportion of the total volume loss. Zhang et al. [248] introduced a metal Zr layer into the ZrN coating to prepare a Zr/ZrN multilayer coating on the surface of titanium alloy, reducing the cylindrical grain boundaries and growth defects of the coating. The Zr layer in the multilayer coating could preferentially form a ZrO_2_ passivated film, and the multilayer structure could effectively inhibit the penetration of corrosive media so that it had good corrosion resistance. At the same time, the introduction of Zr metal layer inhibited the crack growth during the wear process and further improved the tribocorrosion resistance of Zr/ZrN multilayer coating. This study revealed the penetration law of corrosive media in the multi-layer coating and the formation mechanism of passivation film on the surface of titanium alloy and clarified the interaction mechanism between corrosive media and cracks in the process of corrosion wear, which can provide a theoretical basis for the further application of multi-layer coating in the surface protection of medical titanium alloy.

Diamond-like carbon (DLC) coatings are known to be excellent candidates for using as protective coatings on biomedical alloys, not only due to their excellent tribological properties but also due to their chemical composition and stability [249,250,251,252,253,254]. Hinüber et al. [255] fabricated a ta-C DLC on Ti6Al4V alloy, which presented a good adhesion strength, together with reduced tribocorrosion rate in PBS solution and excellent in vitro biocompatibility. E. Arslan et al. [256] investigated the tribocorrosion behavior of the Ti6Al4V alloy and the Ti-DLC coating deposited on the Ti6Al4V substrate using closed-field unbalanced magnetron sputtering (CFUBMS) under tribocorrosive conditions in sliding contacts. The open-circuit potentials were affected by rubbing and changed significantly for the Ti6Al4V substrate and the Ti-DLC coating during rubbing. The tribocorrosion performance of the Ti-DLC coating was better than that of the Ti6Al4V substrate. DLC coatings demonstrated a high protection efficiency (about 97%) against corrosion of the Ti6Al4V substrate, drastically reducing the corrosion current density in static conditions compared to the Ti6Al4V alloy [257]. The tribological properties were significantly improved under dry conditions. However, the life of DLC films was reduced two to ten times in the tribocorrosion tests by the simultaneous action of tribocorrosion mechanisms. 

At present, hard coatings are usually deposited on the surface of titanium alloy using vapor deposition technology to improve the friction and wear properties of titanium alloy substrates. However, due to the significant differences in hardness, thermal expansion coefficients, and elastic moduli between the hard coating and the titanium substrate, as well as the high residual stress inside the coating, the bonding strength between the coating and the substrate is low, and even brittle cracking and shedding occur. In addition, for single-layer binary and multi-element hard coatings, with the extension of deposition time and the increase in temperature, the grains gradually coarsen and form columnar crystals, resulting in intergranular cracks. Structural defects such as grain boundaries and intergranular microcracks are often fast channels for aggressive ion permeation of coatings. 

### 5.4. Laser Cladding

Laser cladding refers to adding external materials to the molten pool formed with laser irradiation of the substrate, employing synchronous or preset materials and rapid solidification of the two to create a cladding layer [258,259,260,261,262]. The characteristics of laser cladding include: the cladding layer has low dilution but strong bonding force and is metallurgically bonded to the substrate, which can significantly improve the wear resistance, corrosion resistance, heat resistance, oxidation resistance, and electrical characteristics of the substrate material surface to achieve the purpose of surface modification or repair, meet the specific performance requirements of the material surface, and save a lot of material costs [263,264,265,266,267,268]. The process parameters of laser cladding mainly include laser power, spot diameter, cladding speed, defocusing amount, powder feeding speed, scanning speed, preheating temperature, etc. These parameters significantly influence the dilution rate, cracking, surface roughness of the cladding layer, and compactness of the cladding parts. The parameters also affect each other, which is a very complex process. It is necessary to adopt reasonable control methods to control these parameters within the allowable range of the laser cladding process.

Feng et al. [37] prepared a new Ti-Al-(C, N) composite coating on TC4 substrate using laser cladding technology. Then, the electrochemical properties of the mixed layer under static and dynamic conditions were analyzed in an artificial seawater environment, and the tribocorrosion behavior and corrosion–wear synergistic mechanism were studied using the tribocorrosion test system. The results show that Ti_2_AlC and Ti_2_AlN self-lubricating phases in the composite coating and the corrosion products with particular lubrication effects produced during the friction process keep the average friction coefficient low. With the increase in load, the mechanical failure of the passive film during the friction process is enhanced, the adsorption of the corrosive medium on the surface of the passive film leads to its active dissolution, and the wear amount increases. The corrosion range of the composite coating in artificial seawater is not extensive, and the wear behavior mainly determines the volume loss caused by friction corrosion.

Obadele et al. [269] used the ytterbium-doped laser system to laser clad commercially pure Ti particles on Ti6Al4V. They also evaluated the tribocorrosion behavior of laser cladding Ti6Al4V on the sliding of alumina balls in 3.5% NaCl. The corrosion potential of the Ti6Al4V-coated sample was lower than that of the uncoated sample. In addition, in the two electrolytes studied, the corrosion current density of the cladding Ti6Al4V was significantly reduced. The tribocorrosion behavior of the cladding Ti6Al4V did not change significantly. The Ti6Al4V cladding layer was greatly reduced in the modified 3.5% NaCl, while it remained unchanged in 3.5% NaCl. In Ti6Al4V, the friction coefficient increased with time, while the friction coefficient of the cladding sample decreased slightly with time. Pejaković et al. [36] developed a 316L laser cladding layer reinforced with titanium carbonitride grains. An agglomerated sintered (Ti, Mo) (C, N) -Ni powder was designed to maintain the chemical integrity of the challenging phase during laser deposition. The tribocorrosion resistance of the sample in artificial seawater was increased by more than ten times under the condition of open circuit potential and by more than thirty times under the condition of passive potential.

Zhang et al. [163] prepared a TiN + TiB/TC4 titanium-based composite coating using laser cladding. The preparation process, characterization of tribocorrosion performance, and wear mech-anism are shown in Figure 15. The results showed that adding BN promoted the in situ formation of microsized elliptical TiN phases and nanosized fibrous TiB phases, forming a new structure with the hard TiN surrounded and anchored by coherent TiB. With increasing quantities of TiN and TiB phases, the material did not undergo significant plastic deformation. During the tribocorrosion process, a considerable accumulation of satisfactory debris, repeated compaction, and relative motion on the surface formed a dense TiO_2_-MoO_3_ oxide film. When exposed to a corrosive chloride ion environment, the thick oxide film transformed into a two-layer structure consisting of a dense inner layer and a porous outer layer, significantly reducing the tribocorrosion rate.

### 5.5. Duplex Treatment

There are many kinds of surface treatment technologies for titanium metal, each with its advantages and disadvantages. It is an inevitable trend to develop a variety of surface treatment technologies and synergistic protection for titanium metal parts used in extreme environments, such as nitriding, ion implantation, laser shock peening and vapor deposition coating composite technology, surface texturing and coating composite technology, micro-arc oxidation, and hybrid treatment technology [249,270,271,272,273,274,275]. Table 3 below shows some of the improvements in tribocorrosion of titanium alloys with duplex treatment.

Kao et al. [281] showed that the duplex nitriding/TiN coating treatment significantly improved the tribological, anti-corrosion, and biocompatibility properties of the original Ti6Al4V alloy in 0.9 wt.% NaCl solution. Li et al. [279] investigated the TiSiCN/nitride duplex coatings prepared using gas nitriding and multi-arc ion plating. The effect of wear on corrosion was significant and the action of corrosion determined the total degree of synergy in the tribocorrosion system. Furthermore, the friction coefficient was greatly reduced through duplex coatings during the tribocorrosion test due to the good resistance to seawater and the graphitization effect produced by the TiSiCN layer. Zammit et al. [282] reported a two-phase treatment consisting of shot peening (SP) and physical vapor deposition (PVD) coatings to improve the surface properties of additive manufactured Ti-6Al-4V material. SP and duplex treatments resulted in a 13% and 210% increase in hardness, respectively. While untreated and SP-treated samples showed similar frictional corrosion behavior, the dual-treated samples showed the greatest resistance to corrosion wear, with no surface damage and a reduced rate of material loss.

Zhang et al. [283] prepared a dual-phase TiN-MAO coating on TC17 alloy using a hybrid treatment method combining MAO and high-power pulsed magnetron sputtering (HiPIMS). The corrosion and tribocorrosion properties of the bare substrate and coating substrate were comprehensively analyzed. The results of tribocorrosion showed that the synergistic effect of tribocorrosion lead to severe wear and adhesive wear of the bare substrate and MAO coating substrate. However, the dual-phase TiN-MAO coating exhibited the best tribocorrosion resistance, achieving the lowest CoF value (about 0.17) and wear rate (about 6.21 × 10^−5^ mm^3^·N^−1^·m^−1^) simultaneously under cathodic protection. The surface hardness of the dual-phase TiN-MAO coating was high, and only slight fracture and delamination occur on the wear scar. The irregular solid particles falling off the surface coating were trapped in the wear trajectory, resulting in third-body abrasive wear. The wear particles fell off and accumulated on the surface of the wear scar, playing the role of a rolling ball. The wear mechanism changed from sliding to rolling friction, significantly reducing the CoF. The corresponding friction mechanism is shown in Figure 16.

The tribocorrosion behavior of a DLC coating system, as obtained by Hatem et al. [284], was investigated through a hybrid technique involving plasma immersion ion deposition (PIID) and plasma-enhanced magnetron sputtering (PEMS). The corresponding results can be seen in Figure 17. In PIID technology, the use of carbonaceous precursors (such as methane or acetylene) can produce DLC coatings with a very high hardness (10–25 GPa) and a thickness of up to 2 μm. Compared with other traditional technologies, the method is simpler and the cost is lower [285]. PEMS technology relies on the sputtering discharge of pure-carbon targets and the decomposition of hydrocarbons by electron impact of precursors [285]. Although PEMS can still deposit DLC, this technology is more suitable for producing carbide and nitride interlayers before DLC + PIID in biomedical coating systems. Therefore, the combination technique can be used to produce a DLC coating system with ideal tribocorrosion behavior because of the higher adhesion between the DLC, interlayer, and substrate. This technique was applied on Ti6Al4V alloy samples, specifically for biomedical applications. Compared with the Ti6Al4V bare alloy sample, the friction coefficient of the coating sample is reduced by at least five times, and the wear rate is less than 2%. The tribocorrosion behavior of DLC coatings prepared using PIID technology and PEMS + PIID technology is related to the mechanical properties and the formation of the carbon transfer layer, and the construction of the carbon transfer layer is closely related to the proportion of sp2 and sp3 bonds in the coating structure. The friction coefficient of the cyclic carbon transfer layer decreases along the reciprocating sliding test under PBS conditions, and the failure elastic strain of the DLC coating has a good correlation with the wear rate. The results show that the appropriate sp2/sp3 bond fraction is crucial to the tribocorrosion properties of DLC coatings.

Kong et al. [286] used a CrN interlayer to deposit a magnetron sputtering cr-diamond-like carbon (DLC) coating layer on a nitrided Ti6Al4V alloy (TAs) surface. Partial results are shown in Figure 18. The results showed that the Cr-DLC coating was an amorphous carbon (C) structure, and Cr clusters were dispersed in the Cr-DLC coating. The crystal structure diagram was used to further analyze the bonding mechanism of the Cr-DLC coating at the interface. After nitriding treatment, some α-Ti was transformed into β-Ti, increasing the TA’s lattice density. The ionized nitrogen bombarded the surface of the nitrided TA to form a nitrided layer, and the dense TiN prevented C from penetrating the nitrided TA during the deposition of the Cr-DLC coating. The nitrided TA provided the load-bearing capacity for the Cr-DLC coating, reduced the hardness difference between the Cr-DLC coating and the nitrided TA, and improved its adhesion. The CrN interlayer connected the Cr-DLC coating with the nitrided TA, which enhanced the adhesion of the Cr-DLC coating at the interface. In this case, the CrN interlayer acted as a high-quality transition layer, which greatly solves the problem of poor adhesion between Cr-DLC and nitrided TA. The first-principles calculations have excellent application prospects for revealing the atomic-level theory of interface bonding.

## 6. Conclusions

In summary, corrosion wear is an essential factor affecting the application of titanium alloy structural materials in variable working conditions. At present, there are still many problems to be solved. There is a large gap between the simulated environment and the complex environment of the actual working conditions, and there are significant limitations in the theoretical guidance of engineering practice. As the equipment’s service environment becomes more complex, the multi-factor coupling condition is more prominent. Given the existing problems of corrosion wear and surface strengthening protection of titanium alloy, the authors believe that conducting in-depth research on the following aspects is necessary.

There are many factors affecting tribocorrosion. The interaction between corruption and wear cannot be superimposed by the loss caused by a single element. It is necessary to conduct damage evaluation device construction, evaluation method establishment, and research revealing the damage mechanism of titanium metal materials under extreme environments, complex working conditions, and multi-factor strong coupling.The traditional research and development system is mostly based on the ‘empirical design–experimental verification’ mode, and the composition and structural design of the auxiliary protective layer are less calculated based on molecular dynamics and first principles. The application of big data engineering, such as high-throughput computing and machine learning in coating research, development, and design, should be strengthened to fully explore the mapping relationship between the composition, organization, and performance of the coating. This will improve the design and development efficiency of high-performance protective layers.Traditional surface treatment technology has many technical difficulties that make it unsuitable for titanium metal treatment. A primary research direction is developing technology and equipment suitable for titanium metal surface treatment. The composite application of various surface treatment technologies, the design and preparation of multi-element, multilayer gradient structures, and ultra-thick surface modification layers meet the surface protection requirements of titanium metal in harsh environments.

## Figures and Tables

**Figure 1 materials-17-00065-f001:**
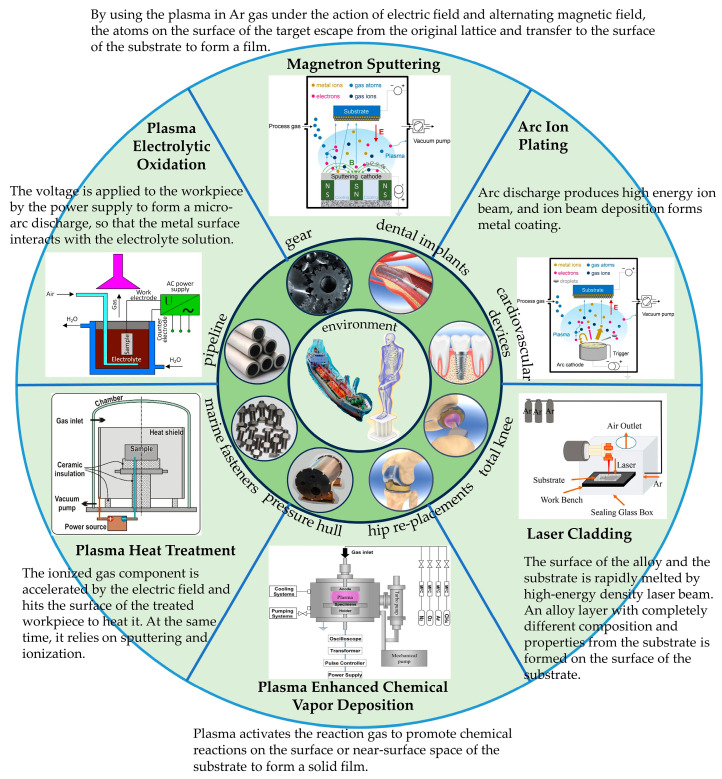
Titanium alloy application scenarios and surface modification techniques and principles for improving the tribocorrosion properties of titanium alloys [55,56,57,58].

**Figure 2 materials-17-00065-f002:**
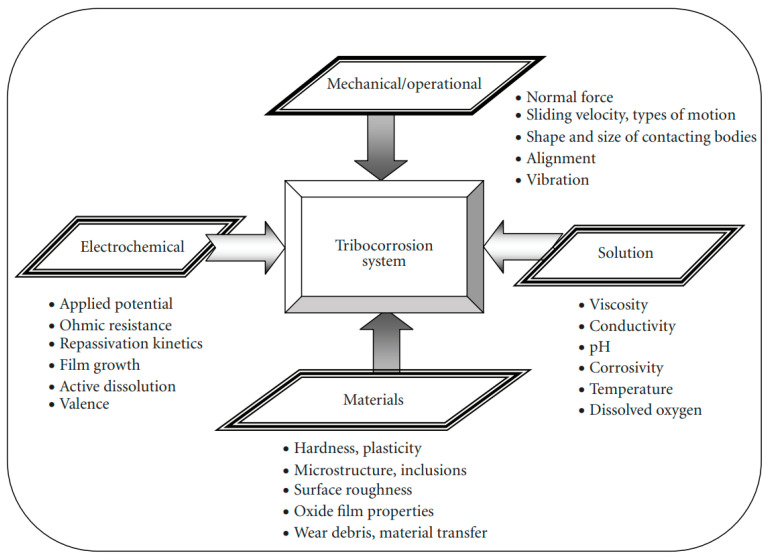
Factors influencing tribocorrosion. Adapted from Ref. [86].

**Figure 3 materials-17-00065-f003:**
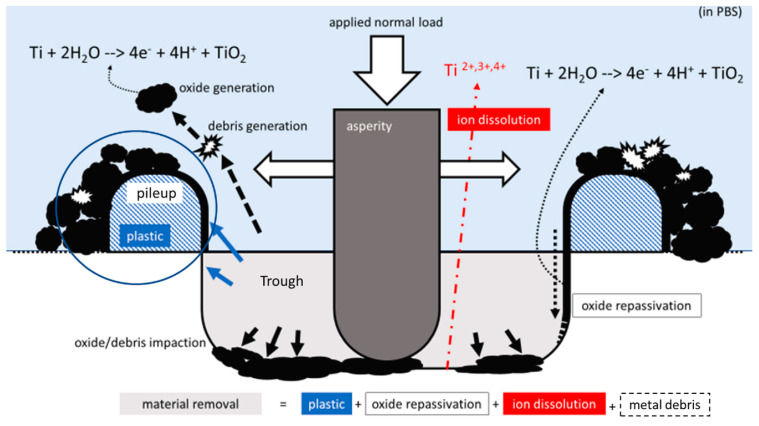
Schematic of the proposed process of tribocorrosion (measured by the volume of material removed) with a single asperity, composed of a combination of tribocorrosion mechanisms (plastic deformation, oxide passivation, and ion dissolution) [87].

**Figure 4 materials-17-00065-f004:**
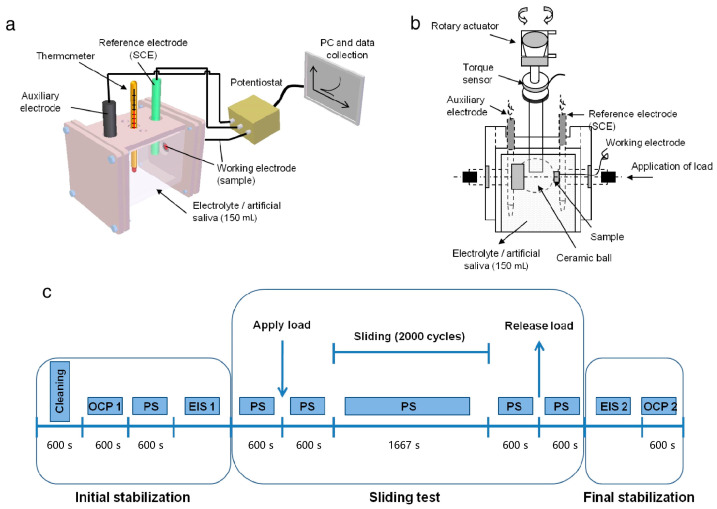
(**a**) Schematic electrochemical setup (a standard 3-electrode cell) used during basic corrosion test. (**b**) Schematic tribocorrosion setup. (**c**) Standard protocol used during tribocorrosion test (OCP—open circuit potential; PS—potentiostatic test; EIS—electrochemical impedance spectroscopy) [88].

**Figure 5 materials-17-00065-f005:**
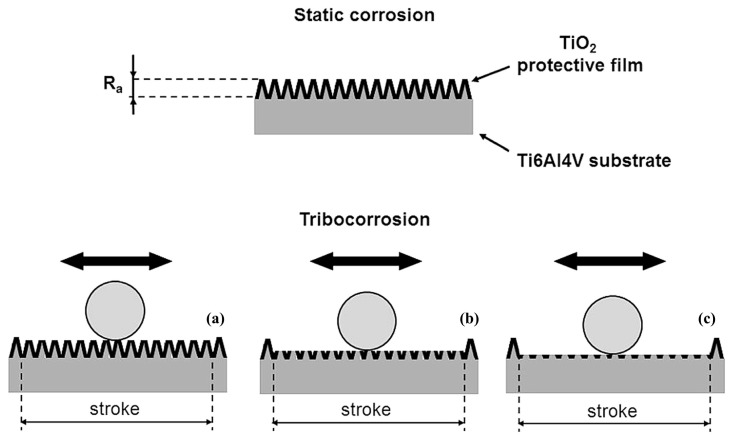
Wear-accelerated corrosion mechanism: (**a**) 10 mN; (**b**) 100 mN; (**c**) 1 N [45].

**Figure 6 materials-17-00065-f006:**
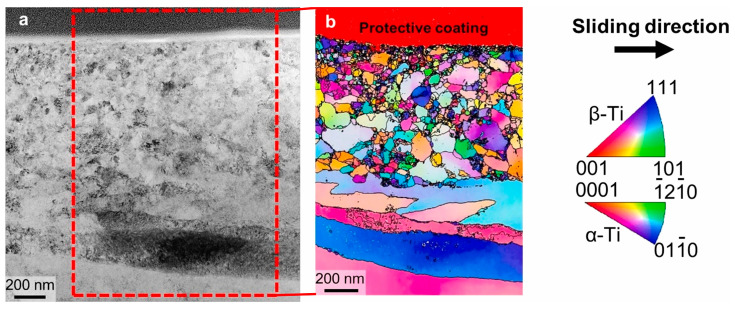
STEM images of an FIB section of the worn surface of the Ti64ELI. (**a**) Bright-field image, (**b**) crystal orientation image using precession electron diffraction [95].

**Figure 7 materials-17-00065-f007:**
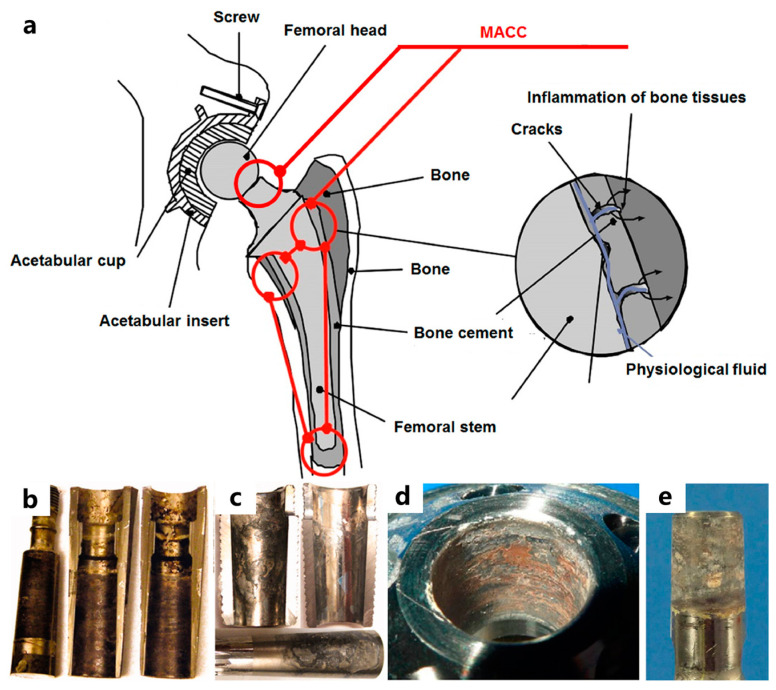
(**a**) Total hip replacement implants [135]; (**b**) lateral male taper and medial half-sleeve female tapers after 22 months implantation, (**c**) proximal female and male tapers after 27 months implantation [136], (**d**) female taper adapter, and (**e**) male stem taper after 43 months implantation [137].

**Figure 8 materials-17-00065-f008:**
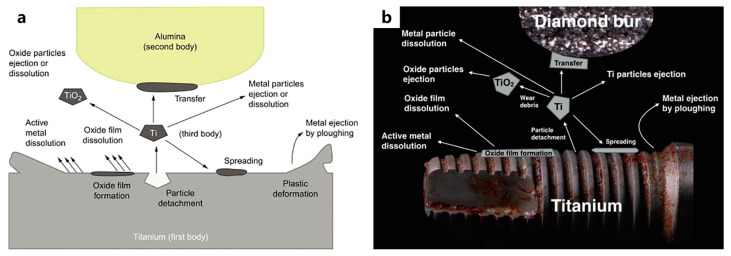
(**a**) Three-body abrasions at the surface of the implant material [140], (**b**) degradation of a titanium implant due to microplowing and active metal dissolution phenomena [141].

**Figure 9 materials-17-00065-f009:**
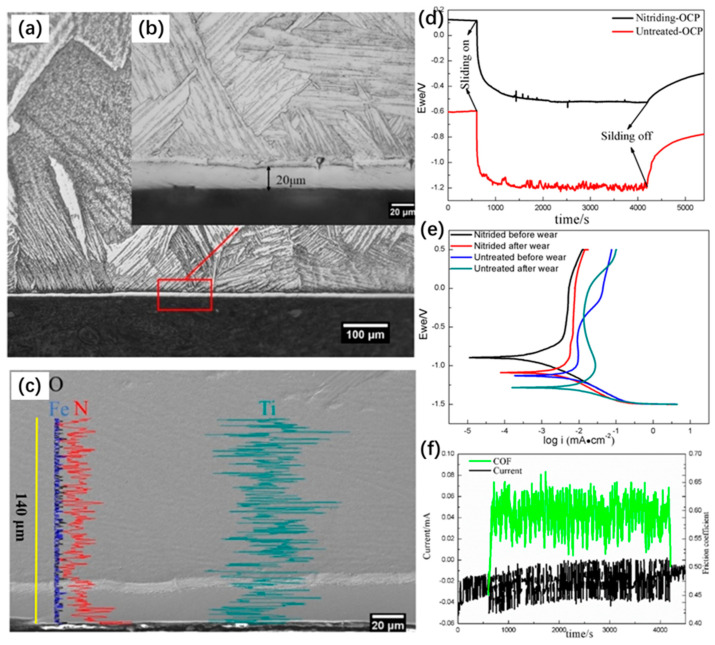
(**a**,**b**) Cross-section metallography of nitriding sample, (**c**) cross-section SEM-EDS of nitriding sample, (**d**) OCP curves of nitriding and untreated sample, (**e**) polarization curve before and after tribocorrosion, (**f**) current and COF of nitrided and untreated sample [194].

**Figure 10 materials-17-00065-f010:**
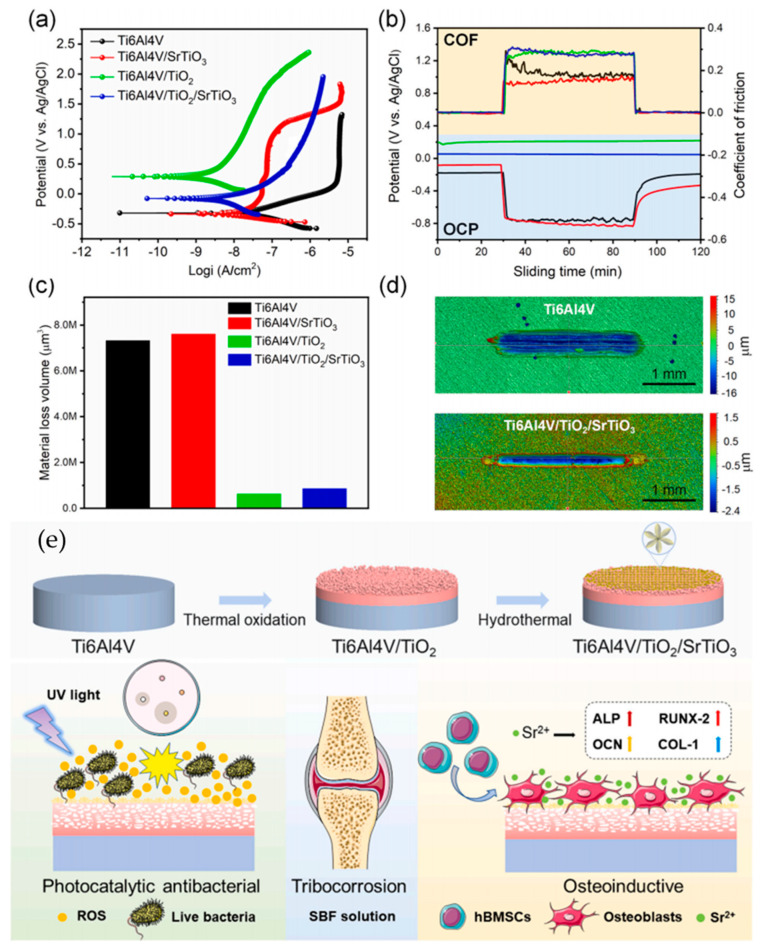
Tribocorrosion characterization of different samples: (**a**) potentiodynamic polarization curves, (**b**) evolution of OCP and COF with sliding time for Ti6Al4V (black), Ti6Al4V/SrTiO_3_ (red), Ti6Al4V/TiO_2_ (green) and Ti6Al4V/TiO_2_/SrTiO_3_ (blue), (**c**) material loss volume, and (**d**) 3D surface morphologies. (**e**) Schematic diagram of the TiO_2_/SrTiO_3_ coating with photocatalytic, antibacterial, osteogenesis, and tribocorrosion resistance properties [201].

**Figure 11 materials-17-00065-f011:**
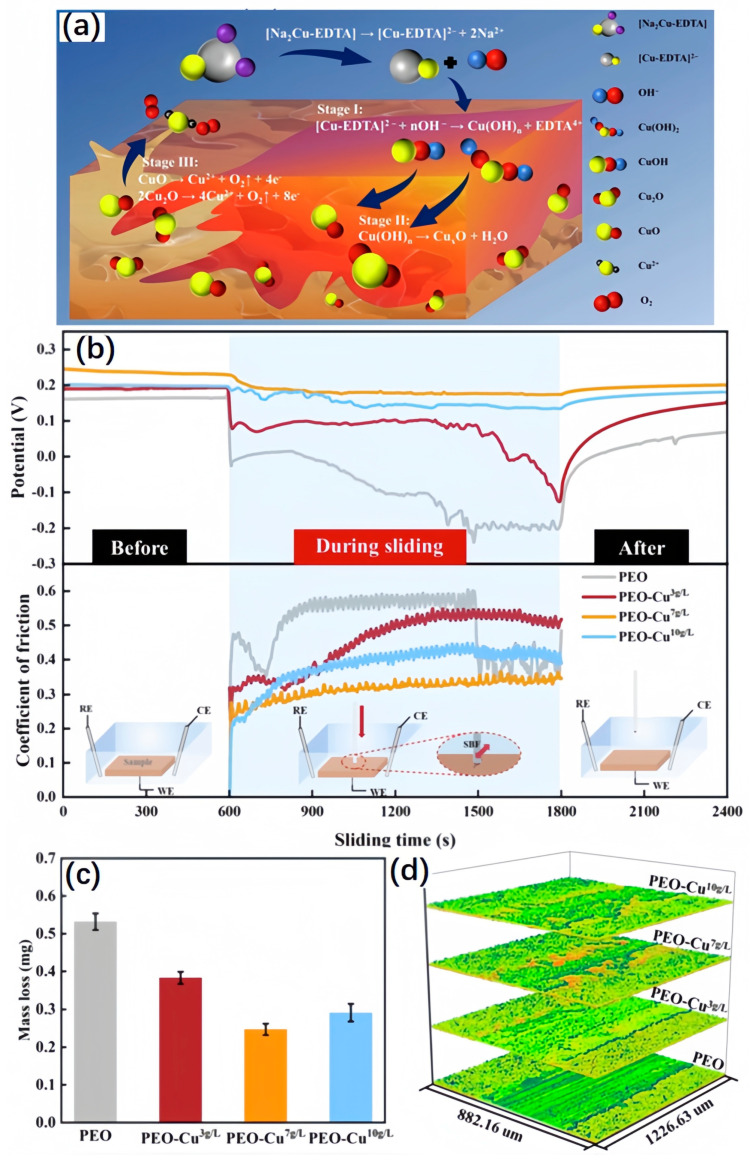
(**a**) Schematic illustration underlying the incorporation mechanism of Cu_x_O phases into the TiO_2_ layer, (**b**) OCP and COF before, during, and after sliding in SBF condition (WE: working electrode, CE: counter electrode, RE: reference electrode), (**c**) the mass loss, (**d**) the 3D topographies [218].

**Figure 12 materials-17-00065-f012:**
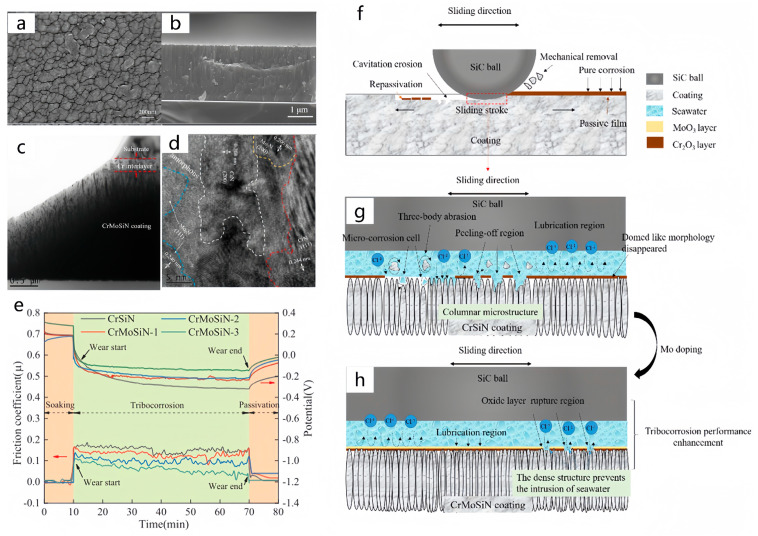
Surface and cross-sectional SEM images of CrMoSiN (**a**,**b**), TEM (**c**), and HRTEM (**d**) images of CrMoSiN coating, (**e**) OCP measurements and respective friction coefficient curves of CrSiN and CrMoSiN coatings, Schematic illustration of tribocorrosion mechanism for Cr(Mo)SiN coatings in artificial seawater (**f**–**h**) [241].

**Figure 13 materials-17-00065-f013:**
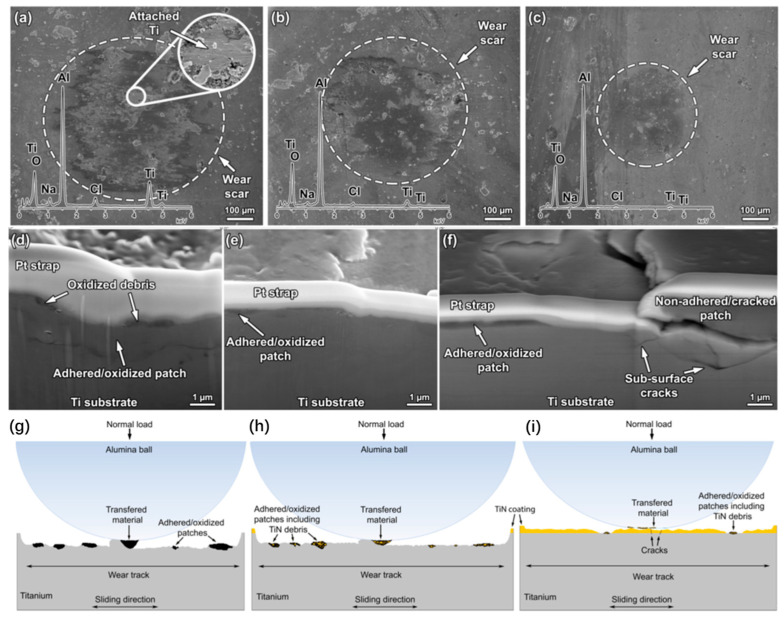
(**a**–**c**) SEM images of worn scars on the surfaces of different samples, (**d**–**f**) magnification SE/SEM images of the worn sub-surfaces of different samples, (**g**–**i**) schematic tribocorrosion mechanisms of different samples [155].

**Figure 14 materials-17-00065-f014:**
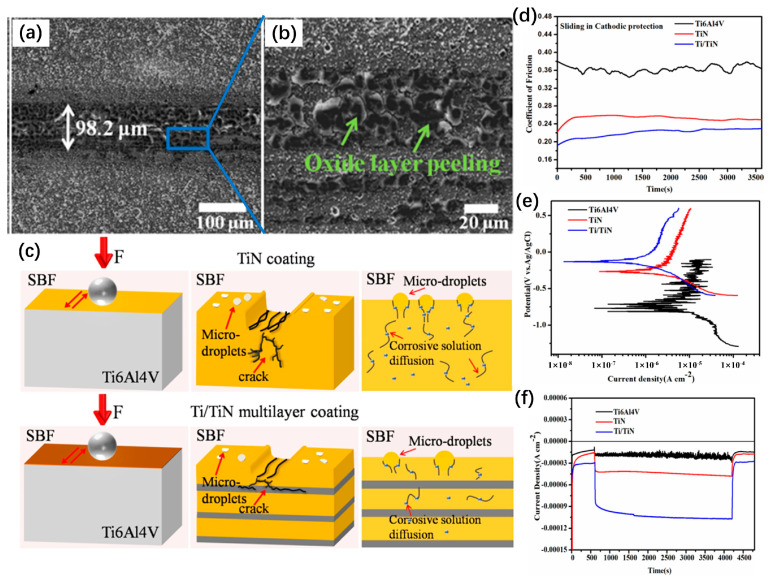
(**a**,**b**) SEM images of wear track sliding, (**c**) wear mechanism models of different coatings in simulated body fluid, (**d**) friction coefficient curves of various coatings, (**e**) potentiodynamic polarization curves of various coatings, (**f**) evolution of OCP values of different coatings [247].

**Figure 15 materials-17-00065-f015:**
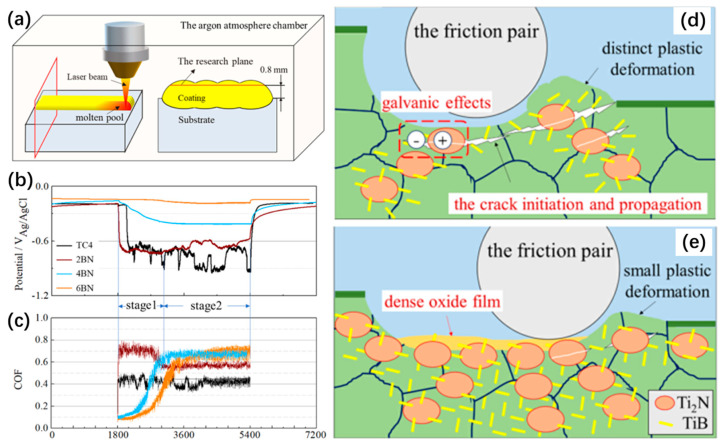
(**a**) The schematic of the laser cladding processes, (**b**) COF, and (**c**) OCP curves of TC4. Mechanism diagram of tribocorrosion process under the different conditions: (**d**) 2 BN, (**e**) 4 BN, and 6 BN coatings [163].

**Figure 16 materials-17-00065-f016:**
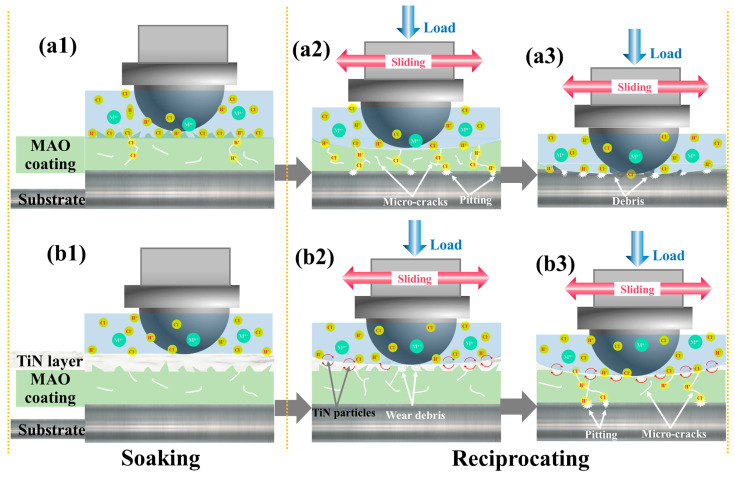
A schematic diagram for wear mechanism of the as-prepared coatings: (**a1**–**a3**) MAO-based coating, (**b1**–**b3**) duplex TiN-MAO coating [283].

**Figure 17 materials-17-00065-f017:**
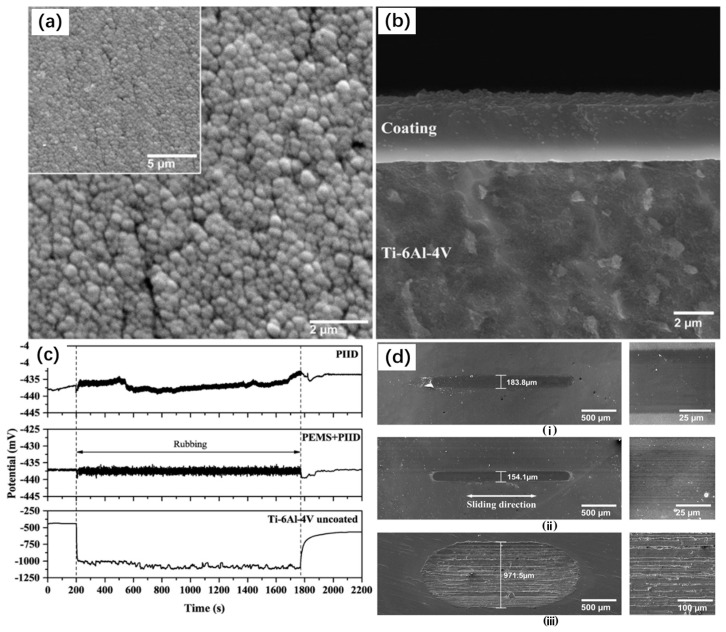
(**a**,**b**) Surface and cross-section SEM images by secondary electrons for the DLC coatings obtained with PEMS + PIID techniques, (**c**) OCP measurements before, during, and after tribocorrosion reciprocal sliding test in 1 × PBS solution for the PIID DLC coating, PEMS + PIID DLC coating, and Ti-6Al-4V uncoated samples, (**d**) SEM images of tribocorrosion wear tracks with measured width at left and center track enlarged at right for the (**i**) PIID DLC coating, (**ii**) PEMS + PIID DLC coating and (**iii**) Ti-6Al-4V uncoated samples [284].

**Figure 18 materials-17-00065-f018:**
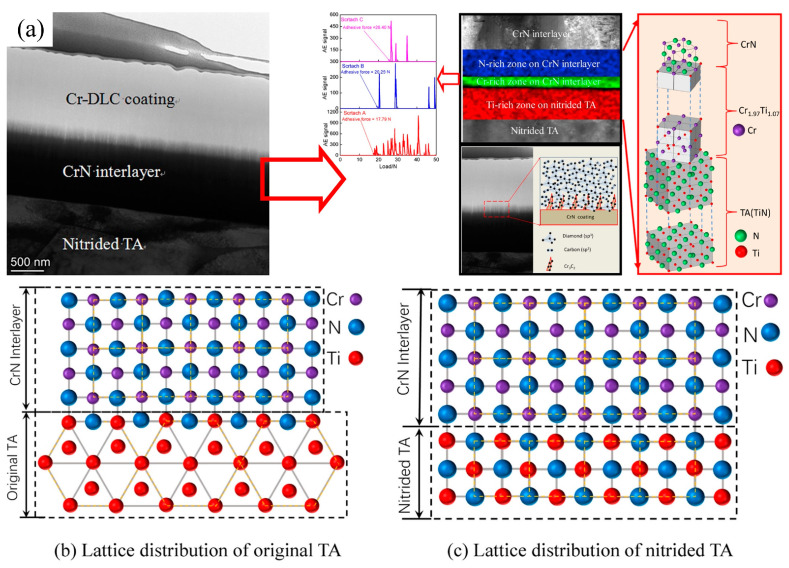
Models of crystal structure, lattice distributions, and adhesive force for the Cr-DLC coating−CrN interlayer−nitrided TA system: (**a**) Model of crystal structure, (**b**) Lattice distribution of original TA, (**c**) Lattice distribution of the nitrided TA [286].

**Table 1 materials-17-00065-t001:** Mechanical properties of titanium and some of its alloys.

Alloy Designation	Microstructure	Elastic Modulus *E* (GPa)	Yield Strength YS (MPa)	Ultimate Strength UTS (MPa)
CpTi	α	105	485	580
Ti-3Al-2.5V	α	118	550	620
Ti-2.5Al-2Zr-1Fe	α	110	570	900
Ti-4Al-2V	α	107	860	960
Ti-4Al-0.005B	α	107	636	710
Ti-10V-2Fe-3Al	metastable β	104	1063	1150
Ti-15V-3Cr-3Sn-3Al	metastable β	80	742	785
Ti-15Mo-3Nb-0.3O	metastable β	82	1020	1020
Ti-35Nb-5Ta-7Zr	metastable β	55	530	590
Ti-35Nb-5Ta-7Zr-0.4O	metastable β	66	976	1010
Ti-6Al-4V	α + β	110	900	970
Ti-6Al-7Nb	α + β	105	921	1024
Ti-6.5V-3.5Mo-1.5Zr-0.3Si	α + β	110	900	1030
Ti-5Al-2.5Fe	α + β	110	921	1024
Ti-13Nb-13Zr	α + β	79	900	1030

**Table 2 materials-17-00065-t002:** Common surface modification methods for improving the tribocorrosion resistance of titanium alloys.

	Materials (Protective Layer)	Surface Treatment	Protective Layer Thickness [μm]	Environment	Counterpart	Load [N]	OCP Values (V)	E_corr_ (V)	I_corr_ (A/cm^2^)	Friction Coefficient	Wear Coefficient [mm^3^/Nm]	Main Wear Mechanism
**Fei Zhou** [147]	Ti6Al4V			aerated artificial seawater	SiC Ball [8 mm]	3 N	−0.8	−0.53	1.86 × 10^−5^	0.23		
	CrMoSiCN coatings	closed-field unbalanced magnetron sputtering	~1				−0.34	−0.30	2.22 × 10^−7^	0.12		three-body abrasion mechanism
**Andre Hatem** [148]	Ti6Al4V			phosphate-buffered saline (PBS) solution	Al_2_O_3_ Ball [6 mm]	10 N	−0.95		7.4 × 10^−6^	0.4	2.61 × 10^−4^	fatigue wear mechanism
	TiSiCN coatings	plasma-enhanced magnetron sputtering	8–10				−0.3		6.8 × 10^−7^	0.27	7.56 × 10^−7^	silicon amorphous restrains the combined action of tribocorrosion
**Lucia Mendizabal** [149]	cp-Ti			PBS solution	Al_2_O_3_ Ball [10 mm]	3 N	−1.07			0.58	4.9 × 10^−8^	
	TaN	magnetron sputtering	2.5				−0.96			0.25	2.4 × 10^−9^	
**Jacek Grabarczyk** [150]	Ti6Al4V			PBS solution	Al_2_O_3_ Ball	10 N	−0.7			0.35	1.18 × 10^−3^	mechanism of adhesive wear
	DLC	magnetron sputtering	1~1.5				−0.1			0.08	4.8 × 10^−6^	combination of adhesive and abrasive wear
**T.M. Manhabosco** [151]	Ti6Al4V			PBS solution	Al_2_O_3_ Ball [5 mm]	4 N	−0.8		6.2 × 10^−7^	0.3		
	TiN, Ti_2_N	chemical heat treatment	~1				−0.4		5.8 × 10^−5^	0.45		
**B. Rahmatian** [152]	Ti6Al4V			PBS solution	Al_2_O_3_ Ball [5 mm]	15 N	−0.75	−0.28		0.33	3 × 10^−4^	
	TiB_2_, TiB	chemical heat treatment	~7				−0.6	−0.1		0.37	1 × 10^−5^	
**Kai yuan Cheng** [153]	Ti6Al4V			bovine-calf serum	Al_2_O_3_ Ball [14 mm]	16 N	−1	0.556	3.8 × 10^−8^	0.45		mechanism of adhesive wear
	TiC, TiO	chemical heat treatment					−0.15	0.466	7.18 × 10^−7^	0.18		lubrication mechanism
**K.M. Li** [154]	Ti6Al4V			HF + HNO_3_ corrosion solution	Al_2_O_3_ Ball [9 mm]	5 N	−1	−0.86	1.1 × 10^−5^	0.5		
	TiN	chemical heat treatment	65				−0.6		4.3 × 10^−6^	0.25		
**I. Çaha** [155]	cp-Ti			0.9 wt.% NaCl	Al_2_O_3_ Ball [10 mm]	1 N	−0.4	−0.371	1.37 × 10^−6^	0.4	5.2 × 10^−7^	combination of abrasive and adhesive wear
	TiN	chemical heat treatment	0.3				−0.3	0.009	7 × 10^−8^	0.6	7.4 × 10^−8^	abrasive and adhesive wear, fatigue wear
**R. Bayón** [156]	Ti6Al4V			PBS solution	Al_2_O_3_ Ball [10 mm]	5 N	−0.9			0.47	0.74	corrosion–abrasion combination
	DLC	arc ion plating	3.03~3.86				−0.06			0.18	4.4 × 10^−4^	
**Yue Wang** [157]	Ti6Al4V			artificial seawater	ZrO_2_ [6 mm]	5 N		−0.27	4.03 × 10^−7^		3.2 × 10^−7^	
	TiSiCN	arc ion plating	2.21~2.47				−0.225	−1.95	3.65 × 10^−7^	0.15	3.08 × 10^−7^	sliding friction transforming to rolling friction
**Yebiao Zhu** [158]	Ti6Al4V			artificial seawater	SiC [6 mm]	5 N		−0.66	9.4 × 10^−6^		2.46 × 10^−4^	
	TiSiN/Ag	arc ion plating	2					−0.33	1.2 × 10^−6^		4 × 10^−6^	
**Minpeng Dong** [159]	Ti6Al4V			artificial seawater	SiC [6 mm]	5 N						
	TiSiCN/Ag	arc ion plating	1.905~2.196				−0.04	−0.19	1.47 × 10^−6^			
**M. Faze** [160]	Ti6Al4V			0.9 wt.% NaCl	SiC [7 mm]	5 N	−0.8	−0.265	3.16 × 10^−8^		6.67 × 10^−8^	
	TiO_2_	PEO					0.3	0.315	1.38 × 10^−8^		2.16 × 10^−8^	
**V. Sáenz de Viteri** [161]	Ti6Al4V			PBS solution	Al_2_O_3_ Ball [10 mm]	3 N	−0.357	0.472				abrasive wear
	Ca,I,P-TiO_2_	PEO	3.1~4.89				−0.001	0.744				adhesive wear
**You Zuo** [162]	Ti6Al4V			3.5 wt.% NaCl	Al_2_O_3_ Ball [6 mm]	2 N	−0.1	−0.25	6.27 × 10^−9^	0.3		
	graphene oxide-TiO_2_	PEO	6.1				0.4	0.35	2.28 × 10^−8^	0.35		
**Hongwei Zhang** [163]	Ti6Al4V			3.5 wt.% NaCl	Si_3_N_4_ [6.35 mm]	5 N	−0.72	−0.9	6 × 10^−5^	0.4		
	TiN + TiB	laser cladding technology					−0.2	−0.6	2.6 × 10^−5^	0.55		

**Table 3 materials-17-00065-t003:** Some duplex treatment methods to improve the tribocorrosion resistance of titanium alloys.

	Environment	Load [N]	Counterpart (Diameter) [mm]	Materials (Protective Layer)	Surface Treatment	OCP Values (V)	E_corr_ (V)	I_corr_ (A/cm^2^)	Friction Coeffcient	Protective Layer Thickness [μm]
**B. Cheraghali** [276]	PBS solution (Reference standards: ASTM F2129 America [277])	1.5	Al_2_O_3_ [8 mm]	CP-Ti			−0.36	7.5 × 10^−8^	0.4	
				oxygen diffusion layers	thermal oxidation		−0.23	2.9 × 10^−8^	0.5	70 ± 5
				TiO_2_	PEO, thermal oxidation		−0.03	2.4 × 10^−9^	0.4	2.3 ± 0.3
**Erfan Abedi Esfahani** [278]	25% foetal bovine serum diluted in PBS electrolyte	5	Ti6Al4V [25 mm]	Ti6Al4V		−0.2	0.15	1.5 × 10^−9^		
			oxygen diffusion layers, TiN [25 mm]	oxygen diffusion layers, TiN	plasma oxidation, plasma nitriding, electron-beam plasma-assisted PVD	−0.1	−0.05	6 × 10^−10^		35~45 diffusion layer 3~6 TiN layer
**Minpeng Dong** [279]	Seawater at 18 ± 3 °C	5	SiC [5 mm]	Ti6Al4V		−0.83	−0.58	1.9 × 10^−5^	0.38	
				TiSiCN	multi-arc ion plating	−0.02	−0.12	1.1 × 10^−6^	0.28	2.8
				TiSiCN, TiN, Ti_2_N	multi-arc ion plating + gas nitriding	0	−0.13	1.7 × 10^−6^	0.25	50 µm diffusion layers
**Chunlei Zhao** [280]	PBS solution (ASTM F2129)	2	SiC [6 mm]	Ti6Al4V		−0.8	−0.6	2.4 × 10^−5^	0.4~0.46	
				TiN	multi-arc ion plating	−0.05	−0.16	1.1 × 10^−6^	0.3~0.32	5.6
				TiN to embed TiO_2_ nanotube coatings	multi-arc ion plating, electrochemical anodization	0.1	−0.02	3 × 10^−7^	0.18~0.26	5.6
**Jacek** **Grabarczyk** [150]	PBS solution (ASTM F2129)	2	Al_2_O_3_ [5 mm]	Ti6Al4V		−0.7			0.3~0.4	
				DLC layers	magnetron sputtering	−0.1			0.1	1~1.5 DLC layers
				oxygen compound layers	plasma oxidation	−0.7			0.4~0.7	
				oxygen compound layers, DLC layers	plasma oxidation, magnetron sputtering	0			0.1	1~1.5 DLC layers
				carbon compound layers	gas carburizing	−0.3			0.5~0.6	
				carbon compound layers, DLC layers	gas carburizing, magnetron sputtering	−0.3~ −0.45			0.1~0.4	0.5~1.5 DLC layers

## Data Availability

The data presented in this study are available on request from the corresponding author.

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
