# Peer review of "Tribocorrosion and Surface Protection Technology of Titanium Alloys: A Review"

_materials, 2023, doi:10.3390/ma17010065_

Round 1

Reviewer 1 Report

Comments and Suggestions for Authors

Reviewer Recommendation and Comments for manuscript materials-2720688-peer-review-v1.pdf with the title: “Tribocorrosion and surface protection technology of titanium alloys: a review”, authors: Yang Li, Zelong Zhou and Yongyong He.

This manuscript is dealing with Tribocorrosion and surface protection technology of titanium alloys in maritime and human environment. With totally 152 literature references, authors review relevant literature in this domain and can be consider to recommend for publishing in MDPI Journal Materials.

The text is clearly written and the graphical interpretation clearly provides insight into the results obtained. The structure, content, and concept of the research work, as well as the achievements, literature resources correspond to the review article. The scientific novelty of this research deals with the development of tribocorrosion and surface protection technology of titanium alloys.  The English is good.

The introduction of manuscript is clear but some improvement can be needed.

The main comments that I find useful for improving the quality of the article are presented below:

 Finding 1: In the introduction, the authors mention the basic characteristics of titanium alloys with reference to the human body, but not to the marine environment. There seems to be a lack of literature sources related to the basic characteristics of titanium alloys as well as for the part in the marine environment.

Finding 2: In the Introduction, the authors do not give the structure of the manuscript and do not clarify state content of the work itself through the upcoming chapters of the work.

Finding 3: Line 40-42. It is missing a reference.

Finding 4: Formula (1) and line 121 Line 40-42. Marks Vtr and V TRit is necessary to equalize.

Finding 5: Formula (3). It is missing clarification of the marks VICPMS.

Reviewer 2 Report

Comments and Suggestions for Authors

The present manuscript aimed to review the tribocorrosion behavior of Ti alloys in different environments and includes different applications. However, the current version still lacks the novelty and direction as compared to other review works. Hence, comments below might be helpful for the revision.

1.      Figure 1 should contain the main map of this paper. It should highlight the main focus of the paper including the application, experimental condition, etc. Thus, the contact lens figure which has no discussion in the text should be eliminated. The sub-topic needs to be rewritten to make it parallel (we believe that Duplex treatment is not parallel with conclusion).

2.      Please rearrange a section about the main characteristics of Ti alloys including microstructural-, structural-, surface-, mechanical-, biological- properties, and degradation behavior. A table might help.

3.      The schematic figure can be reproduced (if necessary), to represent better clarity and quality of the figure. The quality of Figure 2 and 4 needs to be improved. Some figures with small figure annotations need to be reproduced (they are unclear).

4.      The use of Tables and Graphs are very important to show the comparison of one work to the others. The authors may include more tables/graphs. Table 1 environmental section ought to be specified. Thickness of the protective layers should be provided.
